# Study on spatial distribution, regional differences and dynamic evolution of rural financial risk in China

**Wanling Zhou[1], Zhiliang He[2]***

**1** Institute of Finance Engineering in School of Management, Jinan University, Guangzhou, 510632, People's Republic of China, **2** School of Management, Jinan University, Guangzhou, 510632, People's Republic of China

* zhilianghe05@gmail.com

**Data Availability Statement:** All relevant data are within the manuscript and its Supporting Information files.

**Funding:** The author(s) received no specific funding for this work.

## Abstract

Based on panel data from 2009 to 2021, covering 30 provinces in China, we have been constructed the Rural Financial Risk Index using the objective entropy weighting method to study rural financial risk in China systematically from the perspective of spatial distribution. Specifically, we discuss the spatial distribution, regional differences and dynamic evolution of rural financial risk across Chinese four different regions divided into the Northeast, East, Central and West. It's found that *Local government debt* and *Land transfer income* are the two primary determinants influencing the level of rural financial risk in China. Furthermore, we conclude the ranking value of rural financial risk across four regions that the central exhibits the highest level, followed by the West, the East, and finally the Northeast, where the reasons for such ranking results as follows. Firstly, although the highest level of risk among provinces in the West is equivalent to that in the Central, there exists a smaller minimum rural financial risk in the former compared to the latter. Then, it should be noted that there's a low-low agglomeration of rural financial risk in the Northeast, while it demonstrates a high-high agglomeration in the Central according to the Moran Index test analysis. Again, there's a declining trend in rural financial risk disparity within the region and an upward trend is observed when comparing different regions (except the East *vs* West), especially increase largely between the Northeast and Central in past two years after analyzing the decomposition of Dagum Gini coefficient. Moreover, we study the absolute differences and dynamic evolution in different four regions through three-dimensional diagram of kernel density estimation, and it's found that the change of rural financial risk in four regions moved to the right as a whole, while the tail distribution remains inconspicuous. The absolute difference is diminishing in the Northeast, and the two-level differentiation characteristics tend to weaken as a whole in the Central, with a disordered wave peak height observed in both the East and West. Finally, the article presents pertinent policy implications but limitations according to the research findings.

**Competing interests:** The authors have declared that no competing interests exist.

## Introduction

Over the past few years, China's rural financial system reform has achieved remarkable progress. From the property rights reform of rural credit cooperatives to the commercialization of Agricultural Bank of China, and from completing the pilot reform of "agriculture, rural areas and farmers" to developing new rural financial institutions, China's rural financial system is constantly improving and achieving sustainable development [1]. The reform of the rural financial system should prioritize the prevention and resolution of rural financial risks. Enhancing the management of agricultural, rural, and farmer-related financial risks, as well as optimizing the development environment for agricultural finance, have become crucial aspects in studying the rural financial system. Simultaneously, due to the continuous expansion of the rural financial market and ongoing innovation in rural financial products, regional characteristics associated with rural financial risk are increasingly emerging and displaying a trend towards dynamic evolution. In fact, existing literature primarily focuses on exploring the essence, structure, and contagion of regional financial risk to some extent; thus confirming that regional financial risks indeed exhibit genuine spatial spillover effects [2,3]. However, despite being a major agricultural nation in China, the rural financial system still requires further exploration and improvement for the successful implementation of the rural revitalization strategy by rural finance. Examining the impact of regional economy on the rural financial system and studying spatial spillover effects on local financial systems can contribute to stabilizing local economic development. Accurately measuring the correlation degree of local financial risks is crucial for enhancing comprehensive supervision by government authorities and preventing and resolving systemic financial risks [4]. As a major agricultural nation, the robust development of China's rural economy holds immense significance for the advancement of its overall economic growth, with rural finance serving as a catalyst for this progress. However, it is important to acknowledge that while rural finance contributes to economic expansion, it also entails certain risks. Therefore, in order to enhance the contribution of rural finance to rural economic development, it is crucial to measure and analyze the correlation and impact of rural financial risk levels across different regions. Additionally, attention should be given to regional disparities and the evolution of rural financial risks from a spatial perspective, with an aim to strengthen the management of agricultural, rural, and farmer-related financial risks.

In order to investigate the spatial correlation effect of rural financial risk levels and further analyze their spatial distribution, regional heterogeneity, and dynamic evolution characteristics in China, it is necessary to address the following questions: Is there a variation in the level of rural financial risk across different regions in China? If so, what are the underlying mechanisms driving this spatial agglomeration? Considering the disparities in rural economies and financial systems among different provinces and regions in China, does regional heterogeneity exist regarding the level of rural financial risk? What are its dynamic evolution characteristics? Research on these issues holds significant value for enhancing the management system of rural financial risks and improving sustainability as well as stability within rural financial services.

Next, we elaborate the following content arrangement of this paper. The Literature review section presents a comprehensive literature review on rural financial risk, which examines existing theoretical and methodological research in this field to lay the foundation for the current study. The Methods section primarily focuses on describing the research methodology and model principles employed in this paper. In the Empirical research section, a core empirical analysis is conducted to investigate the dynamic evolution process of rural financial risk through measuring the rural financial risk index, analyzing its spatial agglomeration, and exploring regional differences. Finally, concluding remarks are provided in the last section along with relevant policy implications and some limitations.

## Literature review

### Study on rural financial risk

Regional financial risk exhibits strong interconnectivity and transmission capabilities, thereby potentially triggering national financial risks and even global financial crises through chain reactions [5], such as the 2008 financial crisis. Rural financial risk falls within the purview of regional financial risk, which in turn belongs to the "meso" level of financial risk. This distinction sets it apart from micro-level and macro-level risks, making it a distinct manifestation of systemic financial risk. Despite discernible disparities between regional financial risk and micro/macro risks, the former often emerges on the foundation of micro-level risks with contagion and diffusion effects at the macro level [6].

In terms of research on rural financial risk, the majority of studies primarily focus on rural financial institutions and examine the spillover effects of risk at the micro level. With the emergence of new rural financial institutions, some progress has been made in researching risk management for these entities. Du and Hu [7] centered their study around new rural financial institutions and developed a system for investment and financing in rural residential areas. He [8] quantitatively analyzed vulnerability factors that impact the development of rural financial institutions by establishing a relationship model between the ecological environment of rural finance and vulnerability within new rural financial institutions. They also discussed key risks and challenges faced by these institutions. Yu and Cui [9] discovered that new rural financial institutions are particularly sensitive to market conditions due to limited information systems among rural customers, making it challenging for intermediaries to intervene effectively [10]. In pursuit of realizing strategies for revitalizing rural areas, digital inclusive finance has played a crucial role in addressing issues related to exclusion from traditional banking services in these regions. From the perspective of digital inclusive finance, some researches have studied the impact of digital inclusive finance on the risk-taking of rural financial institutions [11,12]. The existing research on rural financial institutions at the micro level has yielded fruitful results in examining risk management of rural finance in China. However, with the implementation of China's regional coordinated and sustainable development strategy, as well as the progress made in China's rural economy and finance, it is imperative to investigate the spatial distribution of rural financial risk from a macro and meso perspective. Therefore, there remains ample room for further research expansion.

In recent years, researchers in rural financial risk management have increasingly recognized the issue of imbalanced regional financial development resulting from the heterogeneity of regional economies. Consequently, this disparity will further contribute to variations in rural financial risk management across different regions of China. Zhao and Zhang [13] pointed out that studying the regional heterogeneity of regional financial risks is of great positive significance for preventing systemic financial risks. Although rural financial institutions address the shortcomings of large commercial banks by supporting agriculture and farmers, they also face a range of risks and hidden dangers. Zhou [14] found that the construction of regional financial risk early warning system and the implementation of regional differentiated credit policies are effective measures to improve regional financial risk management. The study conducted by Li and Wang [15] revealed that differentiated government policies not only address the heterogeneity of regional financial risks resulting from economic disparities among regions, but also cater to the diverse financial needs of farmers within a region. We can be find that the differentiated implementation of government policies is crucial in addressing regional financial risks. Moreover, apart from enhancing external supervision and implementing effective financial oversight measures, it is equally vital to enhance the internal infrastructure of financial institutions [16]. Specifically, it can effectively mitigate regional systemic

financial risks through enhancing the internal infrastructure of financial institutions and implementing external supervisory policies simultaneously [17], which is also applicable to the implementation of rural financial risk management systems.

## Financial risk influencing factors

It's an important research content to study the financial risk management in rural areas, build a risk index system, and analyze the influence of different factors on the risk index. Overall risk management has become a prevailing practice in the risk management of financial institutions, and it is also a breakthrough in the study of regional systemic risk. Systemic risk is commonly evaluated through macroeconomic variables and early warning indicators for micro-level risks on balance sheets, providing a comprehensive reflection of systemic risk [18]. The overall risk of rural finance is not solely determined by micro individual rural financial institutions, but also significantly influenced by the macro environmental background. Zhou et al.'s [19] dynamic CAPM model and risk index system studied China's rural financial risks from both macro and micro perspectives but did not consider regional differences or spatial correlation characteristics of such risks. Previous studies have shown that China's rural financial institution distribution exhibits provincial disparities and regional imbalances [20]. The concentration and diffusion of financial resources result in regional disparities in the financial landscape [21], which consequently give rise to varying degrees of regional financial risks. Therefore, it is crucial to adopt tailored regional policies for rural economic development. Although rural financial institutions are an important part of rural finance, it is inevitable that the study of the overall risk of rural finance only considers the micro individual rural financial institutions, which further encourages the study to consider the change of the overall risk level of rural finance caused by other factors. Only by analyzing the characteristics of regional changes in rural finance can we study the development of rural finance comprehensively and synthetically.

Rural financial risk is a type of regional financial risk. When constructing indicators for rural financial risk, we can draw inspiration from existing literature on the development of indicators for regional financial risks. The existing literature reveals that regional financial risks stem from various factors, ranging from macroeconomic changes to individual micro-level influences, including shifts in the macroeconomic environment, the roles played by banking and insurance markets, government influence, and fluctuations in real estate market. Firstly, changes in the macroeconomic environment can reflect the overall operation of the national or regional economy [22], and it is crucial not to underestimate the influence of the macroeconomic environment on financial risks in rural areas. Jurado et al. [23] confirmed the pivotal role of macroeconomic uncertainty in driving economic business cycles, and its impact on regional risk levels. Then, the stability of the banking and insurance industries, as integral components of financial institutions, is directly correlated with the risk level of the financial market and subsequently impacts regional financial risk [24]. Again, Effective management by local governments promotes sustainable development in local finance [25]. Conversely, if local governments engage in "growth competition," it will lead to an expansion of local debt. Once debt risks emerge, they can propagate systemic public risks through intergovernmental transmission channels and potentially trigger national financial crises and economic downturns [26]. Additionally, fluctuations in real estate market constitute one important source of regional financial risks [27]. The influence exerted by real estate on rural financial risks primarily arises due to its close relationship with land use, excessive real estate development weakens agricultural land utilization. Especially, the emergence of real estate bubble will exacerbate the effects of rural financial risk.

## Assess rural financial risk

The evaluation of overall regional financial risk has been extensively studied through the construction of a risk index model and its integration with machine learning algorithms, which are enabling a comprehensive examination of regional financial risk. For example, Huang et al. [28] performed correlation analysis from regional, financial, and global stock indices, and proposed a model of dynamic topological indicators to measure the systematic risk. The study conducted by Wen et al. [29] employed the Copula function to assess the overall risk level of rural financial institutions based on pertinent data, with a primary focus on market risk, credit risk, and operational risk. However, we consider the regional financial risk influenced by the macro environment, banking and insurance markets, government control, and real estate market, it's necessary to build a risk index model to better study changes of rural financial risk.

How can we comprehensively consider these influencing factors to construct a risk index model? From the previous studies, we find that there are subjective weighting and objective weighting approaches to construct risk evaluation index model. On one hand, The representative methods for subjective empowerment are Analytic Hierarchy Process (AHP) [30,31] and Fuzzy Analytic Hierarchy Process (FAHP) [32,33]. Additionally, FAHP has been compared with AHP by Lee [34], highlighting their respective advantages and disadvantages. Recognizing the limited efficiency and applicability of traditional FAHP, numerous studies have discussed and extended upon the FAHP approach [35,36]. On the other hand, objective weighting methods assign weights to different indicators based on information contained within them. Common objective weighting techniques used for constructing risk indexes primarily include Principal Component Analysis (PCA) [37,38] as well as entropy weight method [39,40].

However, subjective weighting methods often exhibit strong subjectivity due to assigning different weights to specific indicators based on evaluators' preferences, thereby lacking objectivity in evaluation. According to the previous studies, we find that objective weighting methods play a crucial role in constructing regional financial risk index analysis. Therefore, based on the researches of regional financial risk, we choose the rural region as our research perspective, and select relevant risk indicators to analyzes their impact on rural financial risk. It further categorizes these indicators as positive or negative and employs the objective weighting entropy weight method to extract valuable information for constructing a more objective-based rural financial risk index evaluation system.

Moreover, after the construction of the regional risk index model, it is imperative to conduct further research and analysis on the risk index. Yuan (2022) [21] proposed DCN deep learning model to analyze the regional financial risk, such as constructing an early warning model based on the value of the regional financial risk index and expanding the RNN network applied to the construction of the regional financial risk early warning system. Shen et al. [22] constructed the risk index to examine the global and local autocorrelation through the Moran index test, which can be enabling a comprehensive analysis of the spatio-temporal pattern evolution characteristics of China's systemic financial risks and facilitating an investigation into regional risk distribution in China. Liu et al. (2022) [41] analyzed the spatial distribution of financial risks in different regions of China with Moran test, then employed the Dagum Gini coefficient decomposition, and three-dimensional graph of kernel density estimation to examine the relative and absolute differences in financial risk among regions in China. However, from the above researches, we can find that their research perspective was concerned with the situation of China's entire financial market, and did not separately consider the changes of China's rural financial risks in different regions in terms of setting variables. This further shows that there is a large space to expand the research on the regional changes of rural financial risk in China, and this paper fills the gap.

                                                                       

## Discussion

From the previous studies, the increasing trend in research on rural financial risk in recent years, but there remains a significant scope for further exploration in this field. Firstly, see research perspective, the current researches primarily focus on micro-level analysis of risk management within rural financial institutions, with limited studies examining the overall rural financial risk at the meso-macro level. Secondly, although there's consensus regarding the heterogeneity of rural financial risk influenced by regional economies, few investigated spatial correlation and regional heterogeneity of overall rural financial risk from an overarching perspective. Finally, as to research methods, there's a prevalent reliance on relatively singular econometric approaches and subjective weighting when constructing rural financial risk index. Consequently, objective weighting analyses and more methods (such as Moran test, Dagum Gini coefficient decomposition, and kernel density estimation) can be used to better study rural financial risk fully.

Therefore, considering that regional heterogeneity serves as not only the fundamental basis for studying other regional financial risk but also a prominent issue concerning China's rural regional financial risk, we draw inspiration from Liu et al.'s [41] study on China's regional financial risk and Chen and Yang's [20] research on the spatial distribution and diffusion characteristics of China's new rural financial institutions. Firstly, based on the spatial distribution of China's rural economy, we use the entropy weight method in the objective weighting method to construct the rural financial risk index, and also use the ArcGIS software to analyze the change characteristics of geographical spatial diffusion of rural financial risk in China. Secondly, we employ the Moran test to analyze the spatial agglomeration features of rural financial risk in China. Finally, we utilize both Dagum Gini coefficient decomposition and kernel density estimation approach to study the regional disparities and dynamic evolution characteristics of rural financial risk in China. Consequently, our paper provides quantitative support for Chinese different regions in formulating relevant policies aimed at preventing rural financial risk while facilitating accelerated development of China's rural finance.

## Methods

### Rural financial risk index model

To construct the financial risk index of rural areas, it is essential to first select a relevant risk index system. Given that subjective weighting methods are primarily based on subjective evaluation criteria, in order to align with reality and objective circumstances, this paper adopts the entropy weight method as an objective weighting approach to analyze and establish China's rural financial risk index system, as well as synthesize China's rural regional financial risk index. Prior to synthesizing the risk index using the entropy weight method, sample data is standardized. The standardization of data indices positively impacts the promotion of rural financial risk while negatively affecting its inhibition. The entropy weight method calculates information entropy for each index from original data and determines their respective weight coefficients by considering the degree of difference in observed values within composite indices. Higher degrees of difference indicate higher information content and corresponding weights. By effectively eliminating subjective factors, the entropy weight method enables more objective evaluation and analysis of indices, facilitating extraction of valuable information content for synthesizing the rural financial risk index.

The entropy weight method primarily determines the weight of each index by extracting the information entropy associated with that index. Initially, the information entropy $E_j$ of

financial risk indicators in Chinese rural areas is computed, as demonstrated by Formula (1).

$$E_j = -\ln(n)^{-1} \sum_{i=1}^{n} p_{ij} \ln p_{ij}. \tag{1}$$

In Formula (1), $p_{ij} = x_{ij} / \sum_{i=1}^{m} x_{ij}$, $x_{ij}$ represents the value of the $j_{\text{th}}$ indicator of the $i_{\text{th}}$ sample, $p_{ij}$ represents the standardized $x_{ij}$ data index. After calculating the information entropy for each index, the weight $W_j$ of each index can be determined based on the information entropy $E_j$, as shown in Eq (2) below.

$$W_j = \frac{1 - E_j}{k - \sum E_j} j = 1, 2, \ldots, k. \tag{2}$$

According to the weight of each index obtained, the rural financial risk index data of each province in different years can be calculated by combining the original index data.

## Principle of global and local autocorrelation

**Global autocorrelation estimation principle.** In order to investigate the spatial agglomeration of the rural financial risk index, this study employs Moran's test to examine the risk index. Moran's test is an effective method for assessing spatial correlation and agglomeration of financial risk, while global autocorrelation can determine whether there is an overall clustering effect in the spatial distribution of the rural financial risk index. The principle of global autocorrelation is presented in Eq (3).

$$I = \frac{\sum_{i=1}^{n} \sum_{j=1}^{n} W_{ij}(Y_i - \bar{Y})(Y_j - \bar{Y})}{S^2 \sum_{i=1}^{n} \sum_{j=1}^{n} W_{ij}}. \tag{3}$$

In Formula (3), $I$ denotes the value of the global Moran index, and $S^2 = \frac{1}{n} \sum_{i=1}^{n} (Y_i - \bar{Y})$, $\bar{Y} = \frac{1}{n} \sum_{i=1}^{n} Y_i$, where the observed values of the $i_{\text{th}}$ and $j_{\text{th}}$ regions, represented by $Y_i$ and $Y_j$ respectively, are used in conjunction with a spatial weight matrix ($W_{ij}$) to analyze the relationship between adjacent provinces. This paper employs a dichotomous geographical contiguity approach based on the ROOK principle. According to this principle, if two provinces share a common zone or border, they are considered adjacent and assigned a value of 1; otherwise, if there is no shared zone or adjacent public boundary, they are deemed non-adjacent and assigned a value of 0. The expression for the spatial weight matrix $W_{ij}$ is presented in the following Eq (4).

$$W_{ij} = \begin{cases} 1 & (\text{If region } i \text{ is adjacent to region } j) \\ 0 & (\text{If region } i \text{ is not adjacent to region } j) \end{cases}. \tag{4}$$

The Moran index can be considered as the sum-product of observed values for each province, reflecting the spatial distribution pattern of rural financial risk. It ranges from -1 to 1, where a positive value indicates spatial agglomeration and a higher value signifies a stronger degree of agglomeration. Conversely, a negative value suggests dispersion in rural financial risk among regions, with larger values indicating a greater degree of dispersion. A Moran index value of 0 implies random distribution of rural financial risk across regions without any previous correlation.

**Local autocorrelation estimation principle.** If we aim to investigate the presence of a spatial agglomeration effect in a specific region during a particular year, it is necessary to

examine the local autocorrelation of the rural financial risk index. By utilizing measurement software, Moran's local autocorrelation test is conducted on provincial data from that specific year. The principle underlying local autocorrelation is illustrated in Eq (5).

$$I_i = \frac{n(x_i - \bar{x})\sum_{j=1}^{n} W_{ij}(x_j - \bar{x})}{\sum_{i=1}^{n}(x_i - \bar{x})^2}. \tag{5}$$

The value of the local Moran index in region $i$ is represented by $I_i$. Both the spatial weight matrix $W_{ij}$ used for local autocorrelation in Formula (5) and the spatial weight matrix used for global autocorrelation in Formula (3) are geographically adjacent. Local spatial autocorrelation enables the examination of the relationship between the rural financial risk index of a specific area and its surrounding regions, facilitating an assessment of whether there is localized spatial clustering in rural financial risk. Furthermore, Moran scatter plots can be generated to determine positive or negative correlations among provinces within different quadrants. The first and third quadrants indicate positive correlation, while the second and fourth quadrants denote negative correlation. Notably, the first quadrant represents high-high agglomeration, whereas the fourth quadrant signifies low-low agglomeration; similarly, the second quadrant indicates low-high agglomeration, while the third quadrant denotes high-low agglomeration.

## Dagum Gini coefficient and kernel density estimation

**Dagum Gini coefficient and its decomposition.** The traditional imbalance measurement indexes, such as the Theil index and the classic Gini coefficient, have limitations in capturing the overlap between grouped samples and decomposing multiple sub-indexes [41]. However, the Dagum Gini coefficient decomposition allows for a decomposition of the overall difference in samples into within-group differences, between-group differences, and between-group hypervariable density. Specifically, the expression for the overall Gini coefficient $G$ is as follows.

$$G = \sum_{j=1}^{k}\sum_{h=1}^{k}\sum_{i=1}^{n_j}\sum_{r=1}^{n_h} \frac{|y_{ji} - y_{hr}|}{2n^2\bar{y}}, \tag{6}$$

In Eq (6), $n$ represents the number of provinces, $k$ denotes the number of regional divisions, $n_j$ and $n_h$ respectively indicate the number of provinces in regions $j$ and $h$. $y_{ji}$ and $y_{hr}$ represent the rural financial risk index of any province in regions $j$ and $h$ correspondingly, while $\bar{y}$ signifies the average financial risk index across all provinces in China.

When decomposing the Gini coefficient, it is necessary to initially rank the average value of the rural financial risk index within a specific region. Let $\bar{y}_h$, $\bar{y}_j$, and $\bar{y}_k$ represent the mean values of rural financial risks in areas $h$, $j$, and $k$ respectively. It can be assumed that these mean values are arranged in ascending order as follows:

$$\bar{y}_h \leq \cdots \leq \bar{y}_j \leq \cdots \leq \bar{y}_k, \tag{7}$$

Subsequently, we can decompose the Gini coefficient $G$ into Eq (8).

$$G = G_W + G_{nb} + G_t. \tag{8}$$

The variable $G_W$ represents the intra-regional disparity in the rural financial risk index for regions $j$ and $h$. The variable $G_{nb}$ represents the inter-regional difference in the rural financial risk index between regions $j$ and $h$. Lastly, $G_t$ denotes the hypervariable density, which captures any remaining cross-influence of the rural financial risk index between regions. The equations

for $G_W$, $G_{nb}$ and $G_t$ are provided below.

$$G_w = \sum\nolimits_{j=1}^{k} G_{ij} p_j s_j, \tag{9}$$

$$G_{nb} = \sum\nolimits_{j=2}^{k} \sum\nolimits_{h=1}^{j-1} G_{jh}(p_j s_h + p_h s_j) D_{jh}, \tag{10}$$

$$G_i = \sum_{j=2}^{k} \sum_{h=1}^{j-1} G_{jh}(p_j s_h + p_h s_j)(1 - D_{jh}). \tag{11}$$

Where, $p_j = n_j / n$, $p_j$ represents the proportion of the number of provinces $n_j$ in region $j$ to the sample size $n$, $s_h = n_h \bar{y}_h / n\bar{y}$, and $s_h$ represents the proportion of the level of rural financial risk in region $h$ to the level of rural financial risk in all provinces in the sample. According to the formula $\sum_j \sum_h p_j s_h = 1$, the corresponding weights of the within-group and between-group Gini coefficients are $p_j s_h$. $D_{jh}$ represents the relative impact of the level of rural financial risk between regions $j$ and $h$. The expression of $D_{jh}$ is shown in the following equation.

$$D_{jh} = \frac{d_{jh} - p_{jh}}{d_{jh} + p_{jh}}, \tag{12}$$

$$d_{jh} = \int_0^\infty dF_j(y) \int_0^y (y - x) dF_h(x), \tag{13}$$

$$p_{jh} = \int_0^\infty dF_h(y) \int_0^y (y - x) dF_j(y). \tag{14}$$

Where $F_j$ and $F_h$ represent the cumulative distribution function of the level of rural financial risk in regions $j$ and $h$, respectively; $d_{jh}$ represents the total influence between regions $j$ and $h$, and is all $y_{ji}$-$y_{hr}$>0 in regions $j$ and $h$; The weighted average of sample differences can be used as the estimator for the mathematical expectation after summation, and the weight used for each difference is $1/n_j n_h$. We can be see that $p_{jh}$ represents the hypervariable first-order moment between region $j$ and region $h$, and is all $y_{hr}$-$y_{ji}$>0 in region $j$ and region $h$. Summarized mathematical expectation, the sample weighted average with the same weight as $d_{jh}$ can be used as the estimator. According to Formula (12), it is easy to see that $D_{jh}$ represents the proportion of the net influence $d_{jh}$-$p_{jh}$ between regions to its maximum possible value $d_{jh}$+$p_{jh}$, and also there is $D_{jh} = D_{hj} \in [0,1]$.

## Kernel density estimation

Kernel density estimation is a highly significant non-parametric method for estimating the distribution characteristics of rural financial risk levels using density curves. Analyzing the horizontal position of the kernel density curve for individual sample data allows us to assess the level of rural financial risk. The height and width of the curve's peak reflect the degree of aggregation in rural financial risk levels within an interval, while the number of peaks describes the polarization degree of sample data. The extensibility or tailing behavior of the curve indicates how far apart provinces with high or low levels of rural financial risk are from other provinces. A more pronounced tailing suggests a higher degree of differentiation within regions. By vertically comparing kernel density curves across multiple periods in a given region, we can identify dynamic changes in distribution characteristics related to rural financial risk levels.

Horizontal comparisons between morphology (shape) of kernel density curves in different regions allow us to compare differences in trajectories regarding financial risk levels in rural areas. Specifically, Eq (15) illustrates the principle behind kernel density estimation function.

$$f_j(x_{jj}) = \frac{1}{n_j h} \sum_{i=1}^{n_j} K\left(\frac{x_{ji} - x}{h}\right), \tag{15}$$

The function $f_j(x_{ji})$ represents the kernel density function of random variable $x_{ji}$, $n_j$ represents the sample size of the study, $x_{ji}$ is the independent and identically distributed observations, $x$ is the mean value, and $h$ is the bandwidth. The smaller the bandwidth, the higher the estimation accuracy. The symbol $K(\cdot)$ denotes the kernel function, with commonly used options including the Gaussian and Epanechnikov kernels. The choice of kernel function generally has minimal impact on the estimation results. Therefore, this paper adopts the Gaussian kernel function to estimate the sample data for assessing financial risk in rural areas. The estimation principle of the Gaussian kernel function $K(x_{ji})$ is illustrated in Eq (16).

$$K(x_{ji}) = \frac{1}{\sqrt{2\pi}} \exp\left(-\frac{x_{ji}^2}{2}\right). \tag{16}$$

## Empirical research

### Indicators selection and data pretreatment

**Indicators selection.** The accumulation, agglomeration, and contagion process of financial risk in rural areas are intricately linked to the local economic operation. To comprehensively examine the overall risk of rural finance, it is imperative not only to consider the risk status of regional financial institutions but also to account for the impact from the external environment. After analyzing relative researches [19,25,27,29], this paper aims to extract valuable index information from five key aspects to construct the rural financial risk index, which including *Macroeconomy*, *Banking Market*, *Insurance Market*, *Government Factor*, and *Real Estate Market*:

- *Macroeconomy*. It's a crucial determinant of regional financial risks, particularly in the development of rural finance. The stability of the macroeconomy directly influences the risk profile of the financial market and its capacity to withstand regional risks. We employ some indicators related to rural economic development to gauge the changes of rural financial risk. The selected indicators encompass gross agricultural product ($x_1$), per capita consumption expenditure of rural households ($x_2$), and agriculture-related fixed asset investment ($x_3$).

- *Banking Market*. The operation of rural regional banking system serves as crucial factor for assessing the level of rural financial risk. The rural Commercial Bank, serving as a representative, plays a crucial role in the rural financial market. Its business scope is closely intertwined with the development level of both rural economy and finance. We select pertinent indicators from rural commercial banks to measure the progress in rural financial banking, including agriculture-related loan-to-deposit ratio ($x_4$) and non-performing loan ratio ($x_5$).

- *Insurance Market*. The insurance market plays a vital role in the rural financial market. Furthermore, agricultural insurance serves as a crucial safeguard for rural financial institutions to mitigate financial risk, given its extensive participation from individual stakeholders such as rural residents and agriculture-related enterprises. We choose indicators related to agricultural insurance to reflect the situation of the rural insurance market, which including agricultural insurance premium income ($x_6$), agricultural insurance compensation ($x_7$), and agricultural insurance density ($x_8$).

- *Government Factor*. The economic activities and policy orientation of local governments have a significant impact on the fluctuation of rural financial risk. Local governments play a crucial role in driving local economic development, and their agricultural-related expenditures can effectively stimulate the growth of the agricultural economy, but the growth of local government debt promotes the increase of rural financial risk. The selected indicators for government factors primarily include, government expenditure to revenue ratio ($x_9$), ratio of local government debt burden ($x_{10}$), ratio of people employed in rural areas ($x_{11}$), and local fiscal expenditure on agriculture ($x_{12}$).

- *Real Estate Market*. It's essential not to underestimate the impact of real estate market on regional economic development, particularly in light of the potential risks posed by excessive investment and subsequent bubble effects on rural financial institutions and markets. Therefore, this paper employs two indicators from the real estate market, asset-liability ratio in real estate development ($x_{13}$) and land transfer income from real estate development ($x_{14}$).

To sum up, according to the national conditions of our country, while following the above variable selection principles, we consider the economic significance and availability of the selected variables. This study employs a set of 14 indicators from five different aspects to analyze and construct China's rural regional financial risk index. Furthermore, we elaborate on the rationale behind selecting these 14 indicators and examines their positive or negative correlation with the direction of impact on rural financial risk. Please refer to Table 1 in the report for detailed information.

## Data analysis and pretreatment

This paper selects 14 indicators from five aspects, namely *Macroeconomy*, *Banking Market*, *Insurance Market*, *Government Factor*, and *Real Estate Market*, to extract valuable information for constructing the rural financial risk index. According to the principles of data availability and validity, our study excludes Tibet, Hong Kong, Macao, and Taiwan from the sample data. We have selected panel data from 30 provinces (autonomous regions / municipalities directly under the Central Government) in China spanning from 2009 to 2021. Due to the global financial crisis in 2008, the panel data used in this paper starts from 2009 onwards. In order to further investigate the spatial distribution and regional heterogeneity of rural financial risk in China, we categorize 30 provinces into four regions: eastern, western, central, and northeastern regions based on the characteristics of China's regional economic development and the "*Report on China's Regional Financial Operation*", please see the division below.

- **Eastern region**, which comprises ten provinces (municipalities directly under the Central Government), Beijing, Tianjin, Hebei, Shandong, Shanghai, Jiangsu, Zhejiang, Fujian, Guangdong and Hainan.

- **Central region**, including six provinces, Shanxi, Anhui, Jiangxi, Henan, Hubei and Hunan.

- **Western region**, which comprises eleven provinces (autonomous regions / municipalities directly under the Central Government), Nei Mongol, Guangxi, Sichuan, Chongqing, Guizhou, Yunnan, Shaanxi, Gansu, Qinghai, Ningxia and Xinjiang.

- **Northeast region**, including three provinces, Liaoning, Jilin and Heilongjiang.

In light of the missing and unpublished data in certain years, this paper employs the interpolation method to scientifically fill in the gaps. Furthermore, prior to constructing the rural financial risk index, it is essential to subject the data to quantitative tempering processing in order to eliminate any influence caused by non-uniform units on the construction of the

**Table 1. Selection of indicators for rural financial risk index.**

| First-level indicators | Second-level indicators | Influent direction | Selected reasons |
|---|---|---|---|
| Macro-economy | Gross agricultural product ($x_1$) | Negative | Reflect the pace of rural economic development, as well as the economic dynamism in rural areas. |
| | Per capita consumption expenditure of rural households ($x_2$) | Negative | It reflects the consumption level of rural residents and serves as an external manifestation of the expansion of domestic demand. |
| | Agriculture-related fixed asset investment ($x_3$) | Negative | Reflect the role of local governments in promoting rural economic growth. |
| Banking Market | Agriculture-related loan-to-deposit ratio ($x_4$) | Negative | Reflect the profitability of rural financial institutions and the market conditions. |
| | Agriculture-related non-performing loan ratio ($x_5$) | Positive | It serves as a reflection of the asset quality of rural financial institutions and plays a crucial role in assessing bank fragility and liquidity. |
| Insurance Market | Agricultural insurance premium income ($x_6$) | Negative | Reflect the development profitability of agricultural insurance institutions. |
| | Agricultural insurance compensation ($x_7$) | Positive | Reflect the indemnity capacity of agricultural insurance institutions. |
| | Agricultural insurance density ($x_8$) | Negative | The development of rural insurance industry serves as a crucial indicator, and the sound growth of agricultural insurance contributes to the stable operation of the financial industry in rural areas. |
| Government Factor | Government expenditure to revenue ratio ($x_9$) | Positive | Reflect the risk of local fiscal deficit. |
| | Ratio of local government debt burden ($x_{10}$) | Positive | It's a crucial indicator of local financial risk, which in turn contributes to the escalation of rural financial risk. |
| | Ratio of people employed in rural areas ($x_{11}$) | Negative | Reflect the level of activation in the rural economy. |
| | Local fiscal expenditure on agriculture ($x_{12}$) | Negative | Demonstrate the degree to which fiscal expenditure by local governments influences rural economic development. |
| Real Estate Market | Asset-liability ratio in real estate development ($x_{13}$) | Positive | Reflect the asset bubbles of local real estate enterprises. |
| | Land transfer income from real estate development ($x_{14}$) | Positive | The real estate industry's encroachment on rural land has a detrimental impact on the economic development of agriculture-related industries. |

comprehensive index. Therefore, out of the 14 selected indicators in this study, with exception of $x_4$, $x_5$, $x_9$, $x_{10}$, $x_{11}$ and $x_{13}$, all other indicators are treated using growth rates for standardization purposes. For instance, $x_1$ represents the growth rate of agriculture-related GDP and similar approaches are applied for other indicators. The relevant data can be found in S1 Data as provided by this paper.

In the end, the data analyzed in this study primarily originate from the Wind database, China Statistical Yearbook, China Financial Yearbook, China Fiscal Yearbook, and other organized datasets available online.

## Construction of rural financial risk index

After systematically analyzing rural financial risk indicators, we employ the entropy weight method in the objective weighting method to construct the rural financial risk index, and obtain the annual comprehensive rural financial risk value for each province (refer to the studied 30 provinces in China). This paper utilizes data from 30 provinces spanning from 2009 to 2021, encompassing a total of 14 indicators as samples. We analyze the correlation between standardized data and rural financial risk while applying positive adjustment to indicators ($x_5$, $x_7$, $x_9$, $x_{10}$, $x_{13}$ and $x_{14}$) that contribute to an increased occurrence of rural financial risk. Conversely, we apply negative adjustment to indicators ($x_1$, $x_2$, $x_3$, $x_4$, $x_6$, $x_8$, $x_{11}$ and $x_{12}$) that inhibit

**Table 2. Proportion of weight of rural financial risk indicators (Unit: %).**

| ID | Province | Region | $x_1$ | $x_2$ | $x_3$ | $x_4$ | $x_5$ | $x_6$ | $x_7$ | $x_8$ | $x_9$ | $x_{10}$ | $x_{11}$ | $x_{12}$ | $x_{13}$ | $x_{14}$ |
|----|----------|--------|-------|-------|-------|-------|-------|-------|-------|-------|-------|----------|----------|----------|----------|----------|
| 1 | Anhui | Central | 4.85 | 2.66 | 7.50 | 2.31 | 17.40 | 2.18 | 13.11 | 2.19 | 5.50 | 15.39 | 8.48 | 2.28 | 7.16 | 8.99 |
| 2 | Beijing | East | 6.88 | 4.11 | 2.57 | 6.13 | 13.35 | 4.69 | 7.24 | 6.26 | 5.24 | 14.31 | 5.58 | 4.10 | 8.94 | 10.62 |
| 3 | Fujian | East | 5.08 | 2.80 | 8.20 | 6.79 | 12.89 | 2.92 | 7.19 | 3.21 | 4.93 | 16.27 | 6.42 | 5.22 | 5.95 | 12.15 |
| 4 | Gansu | West | 6.10 | 4.18 | 2.57 | 6.91 | 13.70 | 2.76 | 13.07 | 2.78 | 3.49 | 13.88 | 10.89 | 3.97 | 4.46 | 11.24 |
| 5 | Guangdong | East | 5.06 | 5.52 | 4.35 | 3.91 | 12.29 | 2.97 | 12.64 | 3.01 | 11.17 | 15.70 | 4.71 | 6.04 | 4.85 | 7.80 |
| 6 | Guangxi | West | 3.44 | 3.61 | 3.87 | 10.16 | 8.44 | 3.85 | 14.29 | 3.79 | 4.38 | 17.29 | 6.50 | 3.42 | 7.93 | 9.05 |
| 7 | Guizhou | West | 4.62 | 3.93 | 9.75 | 6.23 | 11.07 | 4.10 | 4.37 | 4.20 | 5.75 | 15.85 | 6.58 | 3.56 | 5.29 | 14.71 |
| 8 | Hainan | East | 2.21 | 2.26 | 6.29 | 2.37 | 19.37 | 2.21 | 7.89 | 2.23 | 2.29 | 13.80 | 3.69 | 1.71 | 3.11 | 30.58 |
| 9 | Hebei | East | 6.78 | 3.43 | 3.64 | 4.67 | 10.28 | 2.82 | 9.64 | 3.05 | 5.72 | 19.00 | 8.26 | 2.68 | 2.95 | 17.08 |
| 10 | Henan | Central | 4.84 | 3.09 | 4.39 | 7.57 | 13.94 | 4.31 | 5.84 | 4.34 | 5.78 | 19.83 | 8.33 | 2.74 | 4.40 | 10.62 |
| 11 | Heilongjiang | Northeast | 3.31 | 4.83 | 3.67 | 8.70 | 7.02 | 3.71 | 11.10 | 4.29 | 4.07 | 13.43 | 4.53 | 2.36 | 4.99 | 24.01 |
| 12 | Hubei | Central | 6.58 | 3.55 | 9.61 | 8.57 | 9.54 | 3.71 | 6.33 | 3.75 | 4.57 | 17.44 | 7.36 | 3.41 | 5.50 | 10.08 |
| 13 | Hunan | Central | 8.68 | 2.32 | 5.85 | 3.82 | 7.50 | 5.67 | 5.50 | 6.42 | 5.47 | 14.09 | 17.34 | 2.40 | 5.37 | 9.57 |
| 14 | Jilin | Northeast | 4.03 | 1.80 | 4.83 | 2.44 | 6.98 | 3.20 | 20.70 | 3.22 | 3.00 | 10.14 | 7.23 | 2.73 | 3.30 | 26.41 |
| 15 | Jiangsu | East | 3.43 | 6.43 | 4.57 | 6.18 | 6.00 | 3.63 | 6.66 | 3.54 | 7.29 | 15.13 | 5.76 | 3.43 | 7.69 | 20.28 |
| 16 | Jiangxi | Central | 4.22 | 2.55 | 4.51 | 9.17 | 8.54 | 3.43 | 9.07 | 3.33 | 7.52 | 16.23 | 5.19 | 4.43 | 7.24 | 14.59 |
| 17 | Liaoning | Northeast | 5.39 | 4.95 | 4.22 | 3.16 | 13.16 | 1.91 | 20.94 | 1.94 | 8.30 | 10.29 | 4.29 | 2.16 | 5.15 | 14.13 |
| 18 | Nei Mongol | West | 5.16 | 3.10 | 4.74 | 4.70 | 10.94 | 3.05 | 6.16 | 2.96 | 4.58 | 16.06 | 6.08 | 6.58 | 3.39 | 22.49 |
| 19 | Ningxia | West | 3.29 | 5.09 | 6.56 | 3.32 | 10.77 | 3.47 | 14.33 | 3.40 | 5.32 | 12.84 | 2.77 | 2.63 | 4.57 | 21.67 |
| 20 | Qinghai | West | 3.71 | 2.73 | 2.72 | 9.46 | 6.77 | 2.19 | 15.86 | 2.19 | 6.58 | 17.55 | 5.26 | 4.03 | 2.51 | 18.45 |
| 21 | Shandong | East | 6.71 | 2.42 | 5.30 | 8.20 | 7.95 | 2.29 | 9.12 | 2.31 | 3.99 | 15.02 | 4.71 | 2.52 | 4.07 | 25.40 |
| 22 | Shanxi | Central | 3.01 | 4.38 | 8.93 | 5.43 | 13.86 | 2.31 | 8.77 | 2.31 | 4.43 | 16.37 | 7.94 | 2.66 | 3.49 | 16.11 |
| 23 | Shaanxi | West | 4.88 | 3.30 | 2.91 | 11.11 | 12.96 | 3.40 | 4.79 | 3.46 | 5.38 | 15.80 | 7.72 | 3.04 | 6.10 | 15.17 |
| 24 | Shanghai | East | 3.82 | 3.86 | 9.55 | 5.95 | 6.75 | 2.48 | 10.96 | 2.45 | 6.82 | 14.03 | 9.69 | 9.39 | 4.22 | 10.06 |
| 25 | Sichuan | West | 3.74 | 4.86 | 2.10 | 11.51 | 7.62 | 4.15 | 5.63 | 4.33 | 3.73 | 14.86 | 7.53 | 5.44 | 7.07 | 17.43 |
| 26 | Tianjin | East | 9.78 | 4.71 | 5.25 | 5.22 | 5.67 | 3.86 | 20.98 | 4.44 | 4.00 | 9.56 | 7.45 | 2.37 | 8.18 | 8.53 |
| 27 | Xinjiang | West | 2.47 | 4.15 | 4.54 | 6.97 | 7.40 | 5.40 | 8.45 | 5.43 | 3.40 | 16.23 | 6.46 | 8.47 | 4.65 | 15.99 |
| 28 | Yunnan | West | 3.99 | 3.41 | 5.64 | 8.51 | 7.62 | 3.63 | 4.55 | 3.60 | 4.36 | 16.66 | 8.91 | 3.97 | 6.32 | 18.83 |
| 29 | Zhejiang | East | 6.21 | 4.94 | 4.00 | 4.96 | 6.55 | 3.47 | 8.03 | 3.40 | 11.54 | 20.98 | 6.90 | 6.45 | 4.24 | 8.34 |
| 30 | Chongqing | West | 3.51 | 2.54 | 6.71 | 11.44 | 15.84 | 2.62 | 4.54 | 2.68 | 6.36 | 14.01 | 5.66 | 2.65 | 8.09 | 13.36 |

Note: The grey shading highlights the minimum weight allocation of each province among the 14 indicators, while the underline emphasizes the maximum.

occurrence of rural financial risk. Table 2 presents the weights assigned to indicators ($x_1 \sim x_{14}$) used for synthesizing the rural financial risk index using entropy weight method.

The distribution of the maximum and minimum values of the rural financial risk index among the 14 indicators in each province is presented in Table 2, which aims to analyze the primary and ultimate factors influencing rural financial risk. It can be observed from the Table 2 that, except for $x_4$, $x_9$ and $x_{13}$ indicators, all other indicators exhibit varying weight proportions within the rural financial risk index across 30 provinces. Furthermore, our findings reveal that among these provinces, *Local government debt burden* ($x_{10}$) has the most significant impact on the proportion of rural financial risk index (accounting for 15 out of 30), followed by *Land transfer income from real estate development* ($x_{14}$) with a considerable influence (accounting for 9 out of 30). Conversely, *Local fiscal expenditure on agriculture* ($x_{12}$) holds minimal weight within this context (accounting for only 10 out of 30). It is evident that the *Local government debt burden* and *Land transfer income from real estate development*

constitute two crucial factors influencing rural economic development and rural financial risk. It's not arduous to comprehend that the progress of rural regional economy relies heavily on governmental support. When the proportion of government debt becomes substantial, it directly impacts the uncertainty of rural economy, amplifies the repayment burden for individuals residing in rural areas, and consequently escalates financial risks within these regions. Furthermore, real estate development represents an industry characterized by significant investment scale and high risk levels. As land income derived from real estate development increases, it not only exerts a certain negative influence on economic growth in rural areas but also augments economic uncertainty while jeopardizing financial stability due to potential real estate bubbles caused by excessive development.

In order to conduct a more comprehensive comparison and analysis of the rural financial risk levels in the four regions of China, Table 3 presents the average results of the rural financial risk index from 2009 to 2021 in the four regions, northeast, east, central, and west. Considering the temporal dimension, despite a brief decline in rural financial risk levels across all four regions in 2011, it is evident that there has been an overall upward trend in the rural financial risk level (as depicted directly by Fig 1).

Specifically, according to the changes in rural financial risk levels across the four regions presented in Table 3 from 2009 to 2021, it becomes evident that over the course of the 14-year sample investigation period, the central region consistently exhibits the highest level of rural financial risk. Notably, this region experiences maximum risk levels for five years and minimum risk levels for only one year among all four regions. Furthermore, our analysis reveals that the northeast region consistently demonstrates the lowest rural financial risk index values, with a minimum value occurring in eight out of fourteen years and a maximum value occurring in only one year. Fig 1 visually confirms that rural financial risk steadily increases over time in the central region before reaching its peak. In contrast, despite experiencing a significant spike in risk levels from 2017 to 2019, northeastern China maintains its position as having the lowest mean rural financial risk level among all four regions throughout this entire 14-year investigation period.

By disregarding occasional abnormal situations in the risk level within a specific year, we can establish a ranking of changes in the average rural financial risk values across the four regions during the 14-year sample inspection period. The descending order of mean values for rural financial risk is as follows: central > western > eastern > northeast. It should be noted that this ranking differs from Liu et al.'s [41] study on China's regional financial risk values, which ranked them as northeast > central > west > east.

In addition to the inconsistency of our research objects and sample data, we attribute the divergent ranking results to the following potential factors. The minimal level of rural financial risk in Northeast China can be attributed to two main reasons: firstly, the close geographical proximity among provinces and cities facilitates the aggregation of rural financial resources,

**Table 3. Mean level of rural financial risk in the four regions of China.**

| Region | 2009 | 2010 | 2011 | 2012 | 2013 | 2014 | 2015 | 2016 | 2017 | 2018 | 2019 | 2020 | 2021 |
|--------|------|------|------|------|------|------|------|------|------|------|------|------|------|
| Northeast | 0.35 | 0.23 | 0.19 | 0.31 | 0.30 | 0.31 | 0.32 | 0.36 | 0.38 | 0.52 | 0.43 | 0.43 | 0.39 |
| East | 0.36 | 0.29 | 0.24 | 0.30 | 0.29 | 0.35 | 0.41 | 0.41 | 0.47 | 0.48 | 0.50 | 0.56 | 0.53 |
| Central | 0.35 | 0.30 | 0.24 | 0.29 | 0.35 | 0.31 | 0.39 | 0.47 | 0.50 | 0.57 | 0.57 | 0.63 | 0.63 |
| West | 0.39 | 0.28 | 0.17 | 0.29 | 0.24 | 0.30 | 0.41 | 0.48 | 0.52 | 0.53 | 0.57 | 0.60 | 0.56 |

Note: The grey shading highlights the minimum rural financial risk among the four regions in that particular year. The bold horizontal lines emphasize the maximum value.

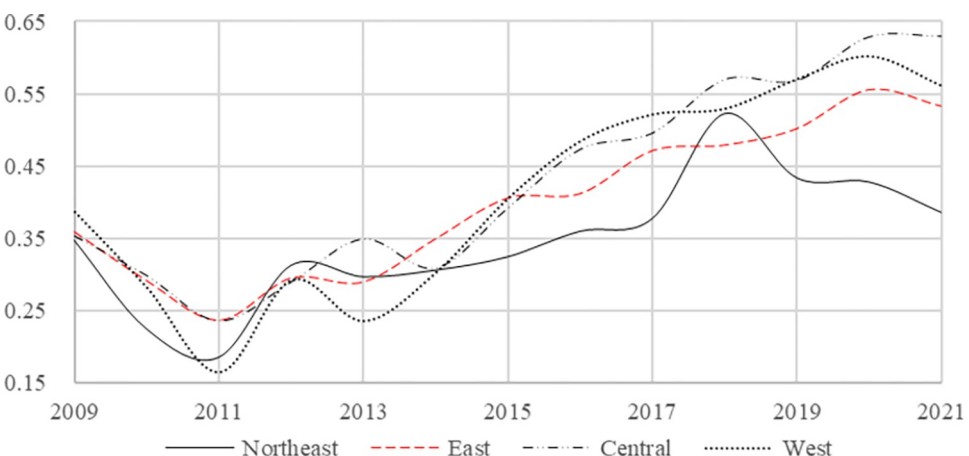

**Fig 1. Time-varying graph of the mean value of rural financial risk in the four regions.**

thereby minimizing information asymmetry and reducing rural financial risk; secondly, Northeast China possesses natural advantages such as fertile black soil and has received substantial support for agricultural development through initiatives like "northeast revitalization". As a result, its agricultural economy operates steadily with a low level of risk in the rural financial system.

Furthermore, regarding the highest mean value of rural financial risk observed in the central region among all four regions, this discrepancy primarily stems from economic structure and policy orientation. Compared with the eastern region which boasts a more advanced development status and diverse economic structure, the central region lags behind with a relatively backward economic structure. Additionally, national policy support for this region is slightly less than that provided to western regions. Many rural areas within the central region have experienced hollowing out phenomena accompanied by significant population loss. Moreover, frequent occurrences of natural disasters in recent years have further heightened risks within its rural financial system.

## Spatial distribution and agglomeration characteristics

In order to visually analyze the spatial distribution and agglomeration characteristics of rural financial risk in 30 provinces across four regions in China, we employ ArcGIS software. It examines the rural financial risk level in China's 30 provinces for the years 2009, 2013, 2017, and 2021 based on the mean value of the rural financial risk. The outcomes are presented in Fig 2, as we can see that the rural financial risk gradually accumulates to the central region as time passes. Specifically, taken together all regions considered that the northeast region exhibits a low level of rural financial risk while both eastern and western regions are about the same with occasional high-risk provinces observed such as Yunnan and Guizhou in the west as well as Zhejiang in the east. Comparatively higher levels of rural financial risk can be found within the central region, indicating a migration trend towards this area at a national scale. Fig 2 intuitively reveals spatial disparities regarding China's rural financial risk levels.

For further studying the level of rural financial risk in 30 provinces across four regions over different years, Table 4 presents the findings regarding the maximum and minimum values of rural financial risk observed in each province within a 14-year sample period. We can be see that the northeastern region of China exhibits the lowest level of rural financial risk. With the exception of Heilongjiang Province (ID: 11) in Northeast China during 2018, where the

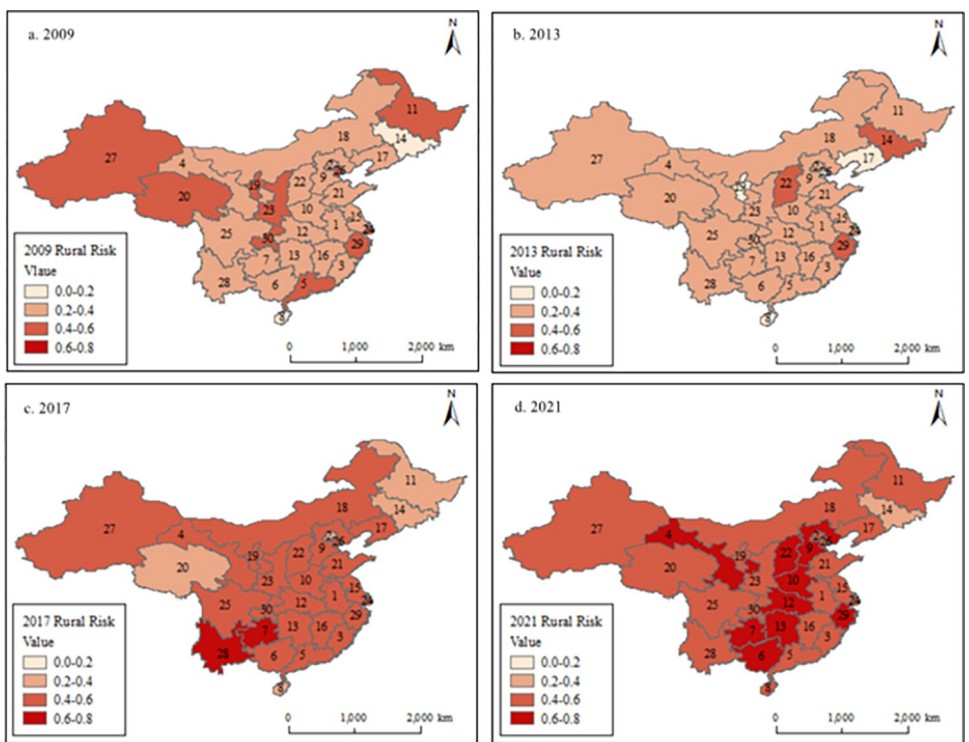

**Fig 2. Spatial distribution and agglomeration of rural financial risk in 2009, 2013, 2017 and 2021.** Note: Because of limiting research data, only 30 Chinese provinces are shown on the picture.

**Table 4. Maximum and minimum values of the rural financial risk level within the four regions.**

| Region | 2009 | 2010 | 2011 | 2012 | 2013 | 2014 | 2015 | 2016 | 2017 | 2018 | 2019 | 2020 | 2021 |
|---|---|---|---|---|---|---|---|---|---|---|---|---|---|
| Northeast | 0.46 | 0.32 | 0.25 | 0.45 | 0.43 | 0.36 | 0.39 | 0.44 | 0.46 | 0.66 | 0.48 | 0.52 | 0.43 |
| | (11) | (11) | (11) | (14) | (14) | (17) | (17) | (11) | (17) | (11) | (11) | (17) | (17) |
| | 0.20 | 0.12 | 0.13 | 0.19 | 0.17 | 0.22 | 0.20 | 0.25 | 0.31 | 0.30 | 0.34 | 0.30 | 0.31 |
| | (14) | (14) | (17) | (17) | (17) | (14) | (14) | (14) | (14) | (14) | (14) | (14) | (14) |
| East | 0.51 | 0.45 | 0.39 | 0.43 | 0.42 | 0.43 | 0.58 | 0.60 | 0.57 | 0.66 | 0.66 | 0.65 | 0.68 |
| | (24) | (8) | (15) | (2) | (29) | (9) | (3) | (3) | (9) | (21) | (29) | (24) | (9) |
| | 0.13 | 0.19 | 0.14 | 0.13 | 0.12 | 0.25 | 0.19 | 0.24 | 0.32 | 0.33 | 0.43 | 0.43 | 0.44 |
| | (8) | (9) | (26) | (8) | (26) | (26) | (8) | (8) | (8) | (8) | (5) | (5) | (21) |
| Central | 0.38 | 0.36 | 0.30 | 0.36 | 0.46 | 0.36 | 0.46 | 0.61 | 0.53 | 0.70 | 0.64 | 0.73 | 0.73 |
| | (22) | (1) | (1) | (13) | (22) | (12) | (22) | (16) | (13) | (16) | (10) | (12) | (12) |
| | 0.34 | 0.25 | 0.15 | 0.24 | 0.30 | 0.27 | 0.36 | 0.38 | 0.41 | 0.43 | 0.52 | 0.55 | 0.54 |
| | (12) | (16) | (13) | (12) | (12) | (1) | (12) | (13) | (22) | (1) | (1) | (1) | (1) |
| West | 0.56 | 0.55 | 0.21 | 0.38 | 0.33 | 0.37 | 0.62 | 0.57 | 0.77 | 0.61 | 0.80 | 0.72 | 0.64 |
| | (30) | (25) | (30) | (20) | (23) | (27) | (23) | (20) | (28) | (28) | (12) | (4) | (4) |
| | 0.22 | 0.13 | 0.12 | 0.20 | 0.15 | 0.24 | 0.32 | 0.37 | 0.37 | 0.44 | 0.42 | 0.48 | 0.48 |
| | (18) | (19) | (28) | (19) | (19) | (20) | (30) | (19) | (20) | (20) | (20) | (25) | (20) |

Note: The gray shading highlights the province with the lowest risk within the region, while the maximum value is emphasized without shading. The values in brackets indicate their respective provinces.

maximum risk level reached 0.66, other years witnessed a fluctuating maximum value of rural financial risk around 0.5. Conversely, the central region experiences the highest level of risk, with its peak value reaching as high as 0.73 between 2009 and 2021.

Specifically, it can be seen from Table 4 that for northeast China, Jilin Province (ID:14) has the smallest rural financial risk value in the three provinces from 2014 to 2021, and its rural financial risk value fluctuates in the range of 0.2–0.3, while Heilongjiang Province (ID:11) and Liaoning Province (ID:17) have larger rural financial risk value. The annual distribution of rural financial risk value in the eastern region is disordered. It can be seen from Table 4 that Hainan Province (ID: 8) has repeatedly appeared in the ranks of the minimum rural financial risk, and Hebei Province (ID: 9) has a higher frequency in the ranks of the maximum rural financial risk. In recent years, the maximum value of rural financial risk in the central region is the largest among the four regions, and its maximum risk value remains high in the range of 0.7, while the minimum value of rural financial risk in the central region also reaches about 0.5. Anhui Province (ID: 1) in the central region has repeatedly appeared in the ranks of the minimum risk, and the six provinces in the central region have taken turns to distribute the maximum value. For the western region, we find that the maximum value of rural financial risk in the western region is no different from that in the central region. However, because the minimum value of rural financial risk in the western region is smaller than that in the central region, the mean value of rural financial risk in the western region is smaller than that in the central region. As can be seen in detail in the Table 4 that Qinghai (ID: 20) and Ningxia (ID: 19) have the lowest rural financial risk distribution in the western region, which reduces the level of rural financial risk.

## Spatial autocorrelation of rural financial risk

**Global spatial autocorrelation.** To conduct a thorough analysis of the spatial clustering characteristics of rural financial risks in China, we have computed the global Moran index value for 30 provinces using the synthesized rural financial risk index data from 2009 to 2021 and geographical adjacency spatial weight matrix data (refer to the S2 Data for further details). Furthermore, the relevant results we can be shown in Table 5.

According to the results of the global Moran's index test, we observe the significant years and find that at significance level of 1%, all significant years except for 2012 showed positive values, while 2012 exhibited a significantly negative value. Furthermore, there has been a clear increasing trend in the global Moran's test results over the past two years, indicating a strengthening spatial positive correlation trend in the financial risk level among rural areas across China's provinces.

**Local spatial autocorrelation.** The global Moran index reflects the overall spatial distribution of rural financial risk level in China, but it cannot identify the spatial correlation and local spatial agglomeration characteristics between each province and other provinces in the

**Table 5. Global Moran index and its significance results.**

| year | Moran's I | year | Moran's I | year | Moran's I |
|------|-----------|------|-----------|------|-----------|
| **2009** | 0.16 | **2014** | 0.00 | **2019** | 0.13 |
| **2010** | 0.10 | **2015** | 0.01 | **2020** | 0.12* |
| **2011** | 0.56*** | **2016** | 0.09 | **2021** | 0.16** |
| **2012** | 0.23 | **2017** | 0.01 | | |
| **2013** | 0.13* | **2018** | 0.04 | | |

Note: ***, **, * indicate significance at the level of 1%, 5% and 10%, respectively.

Table 6. Local Moran's index and its significance results in each province.

| Regions | ID | 2009 | | 2013 | | 2017 | | 2021 | |
|---------|-----|---------|-------|---------|-------|---------|-------|---------|-------|
| | | Moran's I | Quad. | Moran's I | Quad. | Moran's I | Quad. | Moran's I | Quad. |
| Northeast | 11 | -1.41*** | 4 | 0.05* | 1 | 0.54* | 3 | 1.90*** | 3 |
| | 14 | 0.17 | 3 | -1.07* | 4 | 0.31 | 3 | 2.36*** | 3 |
| | 17 | -0.28*** | 4 | -0.55 | 2 | 0.00 | 2 | 0.41 | 3 |
| East | 2 | -0.15 | 2 | 0.28** | 3 | -0.79 | 2 | -0.38 | 2 |
| | 3 | -0.07* | 2 | 1.24** | 1 | -0.09* | 2 | -0.05 | 2 |
| | 5 | -0.30** | 4 | 0.07 | 1 | -0.10 | 4 | -0.24* | 2 |
| | 8 | -0.98 | 2 | -1.00* | 2 | -1.25* | 2 | -0.14 | 4 |
| | 9 | 0.14 | 3 | 0.01 | 3 | -0.17 | 4 | -0.37* | 4 |
| | 15 | -0.05* | 2 | -0.03** | 2 | -0.02 | 4 | 0.07 | 3 |
| | 21 | 0.28 | 3 | -0.01 | 2 | -0.21* | 2 | -0.73* | 2 |
| | 24 | 0.59* | 1 | 0.55** | 1 | -0.16 | 2 | -0.04 | 2 |
| | 26 | -0.62 | 4 | 0.32 | 3 | -0.04 | 4 | -0.20 | 2 |
| | 29 | 0.15* | 1 | 0.97** | 1 | 0.05 | 1 | -0.07 | 4 |
| Central | 1 | 0.02 | 3 | 0.28** | 1 | 0.05 | 1 | -0.04** | 2 |
| | 10 | 0.05 | 3 | 0.45** | 1 | -0.07 | 4 | 0.60** | 1 |
| | 12 | -0.08* | 2 | 0.08* | 1 | -0.03 | 2 | 0.77* | 1 |
| | 13 | -0.04 | 2 | -0.10 | 4 | 0.17* | 1 | 0.53** | 1 |
| | 16 | -0.02 | 2 | 0.39*** | 1 | 0.11* | 1 | 0.24** | 1 |
| | 22 | -0.05 | 4 | 0.43 | 1 | -0.37* | 2 | 0.49** | 1 |
| West | 4 | -0.61** | 2 | 0.32* | 3 | -0.01 | 2 | -0.19 | 4 |
| | 6 | 0.08 | 3 | -0.07 | 2 | 0.37*** | 1 | 0.24 | 1 |
| | 7 | 0.00 | 1 | 0.16 | 3 | 1.03*** | 1 | 0.22 | 1 |
| | 18 | -0.04 | 2 | -0.05 | 2 | -0.50** | 4 | -0.10** | 4 |
| | 19 | -1.24* | 4 | 0.39 | 3 | 0.01 | 1 | -0.09 | 2 |
| | 20 | -0.15 | 4 | 0.25 | 3 | -0.29 | 2 | -0.31 | 2 |
| | 23 | 0.05 | 1 | -0.03 | 4 | -0.03 | 2 | -0.25*** | 2 |
| | 25 | 0.01* | 1 | -0.04* | 4 | 0.13* | 1 | -0.04 | 2 |
| | 27 | -0.06 | 4 | 0.22* | 3 | -0.34* | 4 | 0.07 | 1 |
| | 28 | 0.15 | 3 | 0.34 | 3 | 2.04** | 1 | 0.10 | 1 |
| | 30 | 0.09 | 1 | -0.17 | 2 | -0.10 | 2 | -0.05** | 2 |

Note: ***, ** and * indicate significance at the level of 1%, 5% and 10%, respectively.

four regions. This paper further uses the local Moran index to reveal the local spatial agglomeration characteristics of China's rural financial risk level. Table 6 shows the local Moran index test results of rural financial risk level of 30 provinces in China's four regions in 2009, 2013, 2017 and 2021, and their Moran scatter plot is shown in Fig 3. The distribution of each province in the scatter diagram is the reflection of local correlation. Moran scatter diagram has four quadrants, in which the first and third quadrants represent positive spatial correlation, namely high-high aggregation and low-low aggregation, respectively. While the second and fourth quadrants represent negative spatial correlation, namely low-high aggregation and high-low aggregation, respectively.

According to Table 6, upon comparing the significance results of the four regions in 2009, 2013, 2017, and 2021, it can be observed that the northeast and central regions exhibit an upward trend in their significance results. Furthermore, all these results demonstrate a significant positive correlation, indicating a strong spatial agglomeration characteristic in the

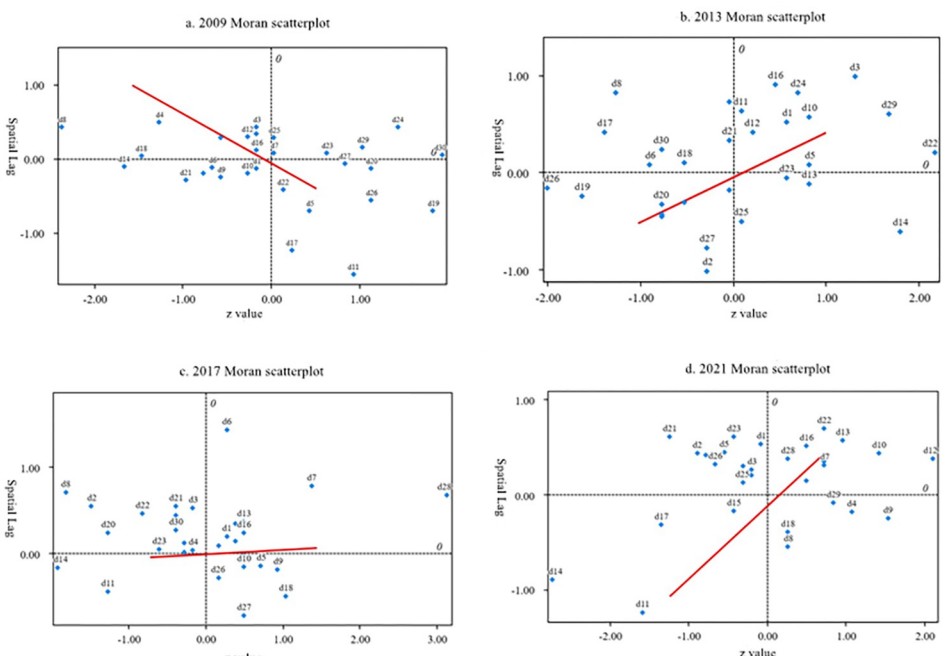

**Fig 3. Quadrant distribution of Moran's index of 30 provinces in 2009, 2013, 2017 and 2021.**

distribution of rural financial risk levels within these regions. Analyzing the outcomes for the eastern and western regions reveals an initial increase followed by a decrease in the number of significant results within each region. Notably, over the past two years, provinces in the eastern region have shown a significantly negative correlation with regards to their rural financial risk levels. In contrast, while there were both significant positive and negative correlations among provinces in the western region during 2017, however, as of 2021 all provinces exhibited significant negative correlations.

In general, the spatial agglomeration characteristics of the northeast and central regions exhibit relative stability. Although there is a positive correlation in the agglomeration trend of rural financial risk levels in these two regions, the northeast region primarily demonstrates a low-low (the third quadrant) agglomeration trend, while provinces in the central region mainly display a high-high (the first quadrant) agglomeration trend. It can be observed that the low-low risk concentration in northeast China results in the lowest rural financial risk among all four regions. Conversely, the high-high risk concentration in the central region leads to the highest mean value of rural financial risk across these four regions.

In recent years, there has been a shift in spatial distribution characteristics of rural financial risk within western China from high-high aggregation to both low-high and high-low aggregations. This explains why at times, the highest value of rural financial risk in western China equals that found within the central region. Compared with eastern China, where agricultural provinces such as Shandong and Hainan are present alongside major economic hubs like Pearl River Delta, Yangtze River Delta, and Beijing-Tianjin-Hebei; it should be noted that spatial distribution agglomeration characteristics for rural financial risks tend to exhibit both low-high (the second quadrant) and high-low (the third quadrant) trends. Consequently, when studying rural financial risks specifically within eastern China's context, one may find less significant results due to unstable agglomeration characteristics, which is one of our paper's

limitations. We consider that estimating spatial agglomeration characteristics for rural financial risks across smaller regional divisions such as economic belts or economic circles would yield more robust empirical findings.

### Regional heterogeneity analysis of rural financial risk

**Decomposition of Dagum Gini coefficient.** In order to conduct a more comprehensive analysis of the rural financial risk across the four regions in China, we employ the Dagum Gini coefficient decomposition to assess both within-group and between-group differences in the rural financial risk values of 30 provinces. The annual calculation results are presented in Table 7, which shows the decomposition outcomes of Gini coefficient decomposition for Northeast (N), East(E), Central(C) and West(W), and the findings on within-group and between-group differences observed from 2009 to 2021 in China's different regions. For a visual representation of overall change, within-group, and between-group differences trends of the Gini coefficient decomposition, please refer to Figs 4 and 5, respectively. Additionally, bar charts illustrating contribution rates for within-group difference, between-group difference, and between-group hypervariable density can be found in Fig 6.

From the analysis of Table 7, it is evident that the Gini coefficient decomposition values in both total and its four regions have exhibited a decreasing trend during the sample investigation period from 2009 to 2021. Thus, it can be inferred that there's a declining trend in regional disparities regarding rural financial risk levels in China. Comparing the total values with regional differences, we observe that China's total Gini coefficient decomposition values surpass those of the central and western regions but falls below that of the northeast region. Prior to 2017, the eastern region had a higher Gini coefficient than total average values, however, after 2017, this reversed. Empirical research findings highlight significant contributions from both the eastern and northeast regions towards variations in rural financial risk levels across China. Notably, within these four regions, it is observed that the central region exhibits the most substantial decrease in financial risk levels during our sample inspection interval.

We examine the between-group differences for rural financial risk in different groups, it becomes evident that the rural financial risk in these regions are characterized by instability, oscillating between periods of decrease and increase. However, when considering 2014 as a dividing point, the disparity in rural financial risk indices between 2014 and 2019 is smaller

**Table 7. Decomposition results of Dagum Gini coefficient of China's rural financial risk.**

| Year | Total | within-group difference | | | | between-group differences | | | | | |
|------|-------|------|------|------|------|---------|---------|---------|---------|---------|---------|
| | | N | E | C | W | N & E | N & C | N & W | E & C | E & W | C & W |
| 2009 | 0.15 | 0.17 | 0.17 | 0.02 | 0.16 | 0.18 | 0.14 | 0.18 | 0.12 | 0.17 | 0.14 |
| 2010 | 0.16 | 0.20 | 0.14 | 0.08 | 0.19 | 0.20 | 0.18 | 0.22 | 0.12 | 0.17 | 0.15 |
| 2011 | 0.17 | 0.14 | 0.19 | 0.13 | 0.08 | 0.20 | 0.18 | 0.13 | 0.18 | 0.21 | 0.20 |
| 2012 | 0.13 | 0.18 | 0.15 | 0.07 | 0.10 | 0.18 | 0.16 | 0.16 | 0.12 | 0.13 | 0.09 |
| 2013 | 0.16 | 0.19 | 0.17 | 0.07 | 0.10 | 0.19 | 0.17 | 0.20 | 0.14 | 0.18 | 0.20 |
| 2014 | 0.10 | 0.10 | 0.10 | 0.05 | 0.09 | 0.12 | 0.09 | 0.10 | 0.10 | 0.11 | 0.08 |
| 2015 | 0.11 | 0.13 | 0.15 | 0.05 | 0.10 | 0.17 | 0.11 | 0.14 | 0.11 | 0.13 | 0.08 |
| 2016 | 0.12 | 0.12 | 0.13 | 0.10 | 0.07 | 0.14 | 0.15 | 0.16 | 0.13 | 0.12 | 0.09 |
| 2017 | 0.10 | 0.09 | 0.09 | 0.05 | 0.10 | 0.14 | 0.14 | 0.17 | 0.08 | 0.10 | 0.08 |
| 2018 | 0.11 | 0.15 | 0.11 | 0.08 | 0.07 | 0.17 | 0.14 | 0.14 | 0.12 | 0.10 | 0.08 |
| 2019 | 0.09 | 0.07 | 0.07 | 0.05 | 0.09 | 0.10 | 0.13 | 0.15 | 0.08 | 0.10 | 0.08 |
| 2020 | 0.09 | 0.11 | 0.07 | 0.06 | 0.07 | 0.14 | 0.19 | 0.17 | 0.08 | 0.08 | 0.06 |
| 2021 | 0.09 | 0.07 | 0.07 | 0.05 | 0.05 | 0.16 | 0.24 | 0.18 | 0.10 | 0.07 | 0.07 |

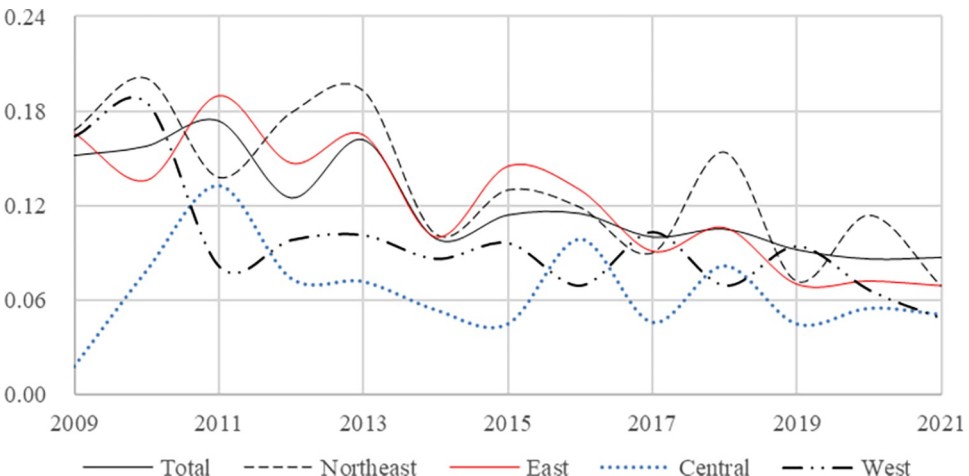

**Fig 4. Trend of differences in Gini decomposition in the total and the four regions.**

compared to that between 2009 and 2014. Post-2019, while there's a downward trend observed in the difference of rural financial risk indices between *E & W*, other groups exhibit an upward trend. Notably, there has been a significant increase in the discrepancy of rural financial risk between *N & C*. Our analysis suggests that national policies favoring development in the western region have resulted in rapid economic growth which has gradually narrowed down economic disparities with the east, consequently impacting economic and financial discrepancies within rural areas as well. Conversely, this growing gap between northeastern and central regions further confirms that on average, the northeast region exhibits lower levels of rural financial risks compared to all other regions whereas the central region experiences relatively higher risks.

Additionally, the between-group hypervariable density reflects the extent to which cross-overlap among samples contributes to the overall difference. The bar chart in Fig 6, depicting the contribution rate decomposed by Gini coefficient, illustrates that the share of contribution

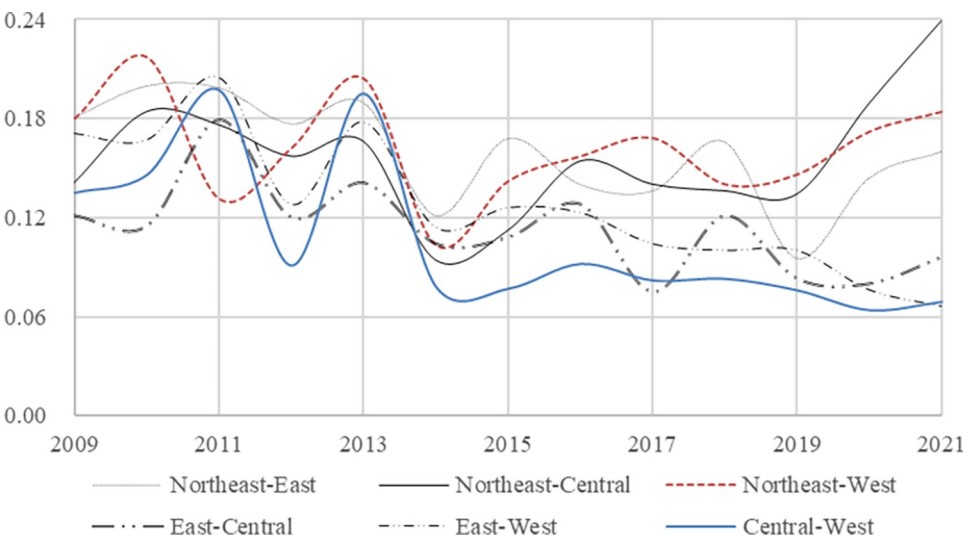

**Fig 5. Trend of differences in Gini decomposition between groups.**

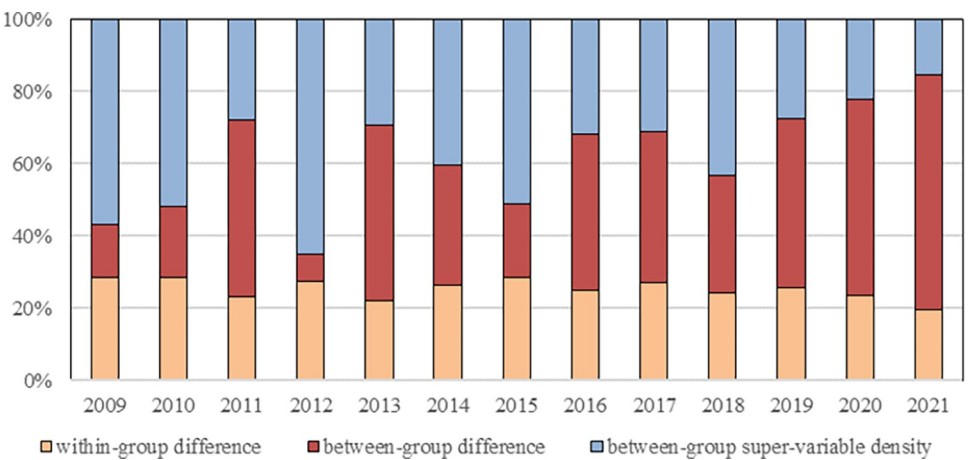

**Fig 6. Spatial differences of rural financial risk level and the results of contribution rate of sources.**

from between-group hypervariable density decreases over time. This indicates that China's regional financial reports are reasonably divided into different regions. Furthermore, recent trends show an increasing between-group difference and a decreasing within-group difference, with the overall difference primarily stemming from the between-group disparity.

## Kernel density estimation and three-dimensional analysis

Although the study of Dagum Gini coefficient decomposition can unveil the value and specific source of the overall disparity in rural financial risk levels in China, as well as identify the changing trajectory of relative disparities across different regions, it fails to depict the time-varying evolution process of absolute disparities among these regions. To reflect the distribution of rural financial risk index in China more comprehensively and capture a better understanding of its dynamic evolution characteristics at a regional level, we employ Gaussian function to derive three-dimensional Kernel density estimation for four regions. In order to visually showcase and compare the dynamic spatial changes in financial risk levels within rural areas of China, our focus lies on key features exhibited by corresponding kernel density curves such as distribution position, peak change form, and distribution malleability. The three-dimensional figure illustrating the dynamic evolution kernel density estimation for rural financial risk across these four regions is presented in Fig 7.

Firstly, through an analysis of the kernel density estimation distribution positions in the four regions, it can be observed from Fig 7 that the overall rural financial risk of these regions exhibits a rightward shift trend during the sample investigation period. Therefore, it is evident that there's an upward trend in the rural financial risk level across these four regions. Secondly, by examining the morphological characteristics of wave peaks, we find that for the northeast region, there were two peaks in 2015 and 2019. Furthermore, in 2019, the peak became higher and narrower in width, indicating a decreasing absolute difference in rural financial risk levels within this region. In contrast, for the central region, there has been a gradual transition from a bimodal state to a single peak over time. This suggests that overall differentiation between two levels within this region tends to weaken and regional disparities are reduced. As for both eastern and western regions, no distinct bimodal distributions are observed. Over time during our sample investigation period though, we notice disordered states with varying heights as well as irregular distributions of highs and lows particularly pronounced within the western region. These findings indicate significant fluctuations in rural financial risk levels among

                                    

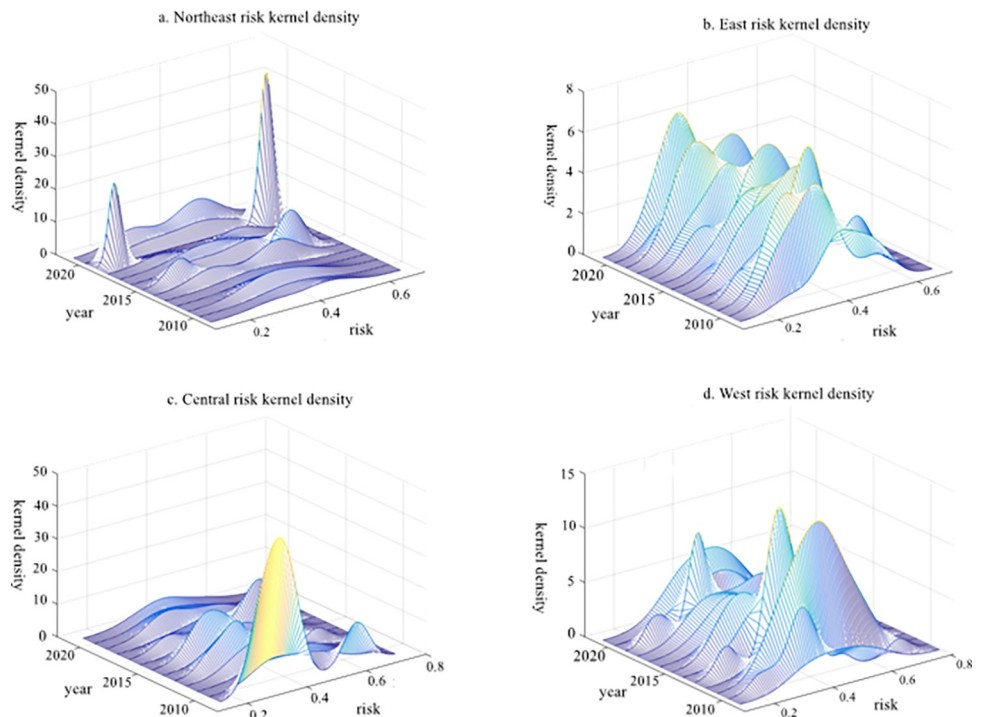

**Fig 7. The three-dimensional dynamic evolution diagram of rural financial risk in four regions.**

provinces within this western area. Finally, regarding tail characteristics of estimated kernel density distributions across all four regions, trailing distributions do not exhibit prominent features.

## Conclusions, policy implications and limitations

### Conclusions

This paper focuses on the annual panel data of 30 provinces in China from 2009 to 2021 as the research subject, and use objective entropy weighting method to construct rural financial risk index for each province. We divide the 30 provinces into four regions (northeast, east, central and west) according to the relevant rules, and examine the variations in rural financial risk levels among these regions. Then, we study the spatial distribution, regional differences, and dynamic evolution characteristics of rural financial risk levels in different regions according to Moran's Index, Dagum Gini coefficient decomposition, and kernel density estimation. Overall, this paper analyzes the changes of China's rural financial risk level in different regions from the spatial perspective, which is a new research perspective. Then, we employ diverse methods to systematically analyze rural financial risk in China, so that making rich content and clear logical structure, which is a valuable attempt and significant contribution in rural financial risk management. The conclusions we can be seen that in the following.

Firstly, we find that the *Local government debt burden* and *Land transfer income from real estate development* hold two crucial factors influencing Chinese rural financial risk after analyzing the five primary indicators and fourteen secondary indicators to construct rural financial risk index among 30 provinces from different four regions.

Secondly, we analyze the spatial distribution of rural financial risk across different four regions in China, it's found that the central exhibits the highest average rural financial risk

value while the northeast displays the lowest. The average value ranking of rural financial risk across four regions in China is as follows, Central > West > East > Northeast. Specifically, we found that the province with highest rural financial risk value in the West, which is no significant difference compared the province with highest in the Central, but the province with minimum value is smaller than the province in the Central, leading to the ranking results.

Then, we study the spatial agglomeration characteristic of rural financial risk among different regions in China and find that these 30 provinces in four regions have a strong spatial correlation, with the trend of positive spatial correlation enhanced gradually in some regions, which further leading to the ranking results of rural financial risk. For details, as we can see that the lowest risk level in the Northeast is associated with low-low agglomeration, whereas the highest risk level in the Central is linked.

Again, we further analyze the relative differences of rural financial risk among different four regions in China though the method of Dagum Gini coefficient decomposition. It's found that the within-group differences show a downward trend in the Chinese rural financial risk in general, especially the Central with the larger decline in the sample period. However, we see the between-group differences exhibit an upward trend for all groups after year 2019 especially the larger upward difference of *N & C*, but except for a downward trend observed between the *E & W*.

In the end, we study the absolute differences of rural financial risk according to three-dimensional diagram of kernel density estimation, and find that the rural financial risk of Chinese four regions shows a trend of moving to the right in general, but the trailing distribution of the four regions is not obvious. According to the peak situation, we can see that it's narrowing gradually for the absolute difference of the Northeast in China, and the two-level differentiation characteristics tend to weaken as a whole in the Central. However, the peak states are unevenly distributed for the East and West, especially we can find that the absolute difference fluctuates greatly for the West.

## Policy implications

We could derive several policy implications according to the research findings presented herein. First of all, greater attention should be devoted to understanding how local government debt and real estate development impact rural economies and contribute to rural financial risks. To enhance prevention and control mechanisms for such risks, it is imperative to establish systems for assessing and monitoring government debt risks as well as disclosing relevant information pertaining to government debts. This will ensure that debt scales remain reasonable while preventing misuse of funds and investment risks associated with debts. Additionally, strict supervision should be implemented regarding land use rights circulation and mortgage within rural areas; rigorous approval processes for rural real estate development projects are necessary measures aimed at curbing excessive utilization of land resources for real estate purposes thereby mitigating potential bubbles or excessive debt-related hazards.

Secondly, it is imperative to enhance the regional disparity in rural financial risk levels, particularly in the western region. It is crucial to address the discrepancies in rural economic and financial development among different provinces by leveraging the synergistic potential of developed provinces with their surrounding counterparts, thereby mitigating the "siphon" effect and reducing polarization. Additionally, attracting foreign capital to less developed provinces should be prioritized while promoting resource and capital spillover from developed provinces. To elevate the level of rural financial risk in the central region, it is essential to amplify the radiation effect of government preferential policies, bolster financial investments in agriculture, and establish a well-developed transportation network within the central region's rural economic belt.

Moreover, considering the relative disparities in rural financial risks among the four regions, it is imperative to implement distinct policies for rural financial support and intervention in risk management. Particular attention should be given to achieving a balanced level of rural financial risks across different regions. The growing relative differences between these regions are closely linked to the uneven distribution of rural financial risks. Therefore, promoting coordinated development of rural finance across all regions is essential. Specifically, concerted efforts must be made in the central region to enhance the local rural financial ecosystem and prevent spillover effects that could potentially impact China's overall rural economic and financial environment. Furthermore, emphasis should also be placed on improving resource allocation efficiency among regions as a whole, fostering integrated urban-rural development, and facilitating high-quality growth in rural areas.

Overall, it is imperative to fully leverage the proactive role of local governments. While promoting rural finance for the facilitation of rural economic development, special attention should be paid to monitoring fluctuations in rural financial risks. Each region's rural economic development necessitates tailored policies that are context-specific. The nation must prioritize regional coordinated development of rural finance and effectively manage the level of rural financial risk, thereby facilitating sustained high-quality growth within China's rural economy and society.

## Limitations

Although this paper has conducted a series of tests and comprehensive research on the spatial distribution, regional heterogeneity and dynamic evolution characteristics of China's rural financial risk level, there are still some limitations. While striving to conduct a normative analysis on the construction of rural financial risk index and indicators selection, some data gaps and differences in research perspectives hindered our efforts. On one hand, due to the complexity of rural financial risk data, it's not entirely perfect about selected indicators and divided not small regions, maybe leading to some insignificant parameters estimation. On the other hand, this paper lacks a more comprehensive spatial econometric method for analyzing causes behind China's rural financial risks—which is also our main focus for future works.

## Supporting information

**S1 Data.**
(XLSX)

**S2 Data.**
(XLSX)

## Acknowledgments

We are grateful to the Editors and three anonymous reviewers for their insightful comments and suggestions.

## Author Contributions

**Conceptualization:** Wanling Zhou.

**Data curation:** Zhiliang He.

**Formal analysis:** Wanling Zhou.

**Methodology:** Zhiliang He.

Writing – original draft: Wanling Zhou.

Writing – review & editing: Wanling Zhou, Zhiliang He.

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
