## [Decision Letter · Decision Letter 0]

12 Jan 2024

PONE-D-23-36724Study on Spatial Distribution, Regional Differences and Dynamic Evolution of Rural Financial Risk in ChinaPLOS ONE

Dear Dr. He,

Thank you for submitting your manuscript to PLOS ONE. After careful consideration, we feel that it has merit but does not fully meet PLOS ONE’s publication criteria as it currently stands. Therefore, we invite you to submit a revised version of the manuscript that addresses the points raised during the review process.

We look forward to receiving your revised manuscript.

Kind regards,

Walid Al-Shaar

Academic Editor

PLOS ONE

Journal Requirements:

Reviewers' comments:

Reviewer's Responses to Questions

**Comments to the Author**

1. Is the manuscript technically sound, and do the data support the conclusions?

Reviewer #1: Yes

Reviewer #2: Partly

Reviewer #3: Partly

2. Has the statistical analysis been performed appropriately and rigorously? 

Reviewer #1: Yes

Reviewer #2: Yes

Reviewer #3: Yes

3. Have the authors made all data underlying the findings in their manuscript fully available?

Reviewer #1: Yes

Reviewer #2: Yes

Reviewer #3: No

4. Is the manuscript presented in an intelligible fashion and written in standard English?

Reviewer #1: Yes

Reviewer #2: No

Reviewer #3: Yes

5. Review Comments to the Author

Reviewer #1: The author(s) have studied“Study on Spatial Distribution, Regional Differences and Dynamic Evolution of Rural Financial Risk in China”. The contents of the research are novel and have potential to be published after some minor comments

1. What is Spatial Distribution?

2. How is this study helpful to the economy of china?

3. What are basic gaps which are filled in this research study?

4. How is this study compared to the previously published? Which issues are addressed in this which are not in the previously?

5. The ranges of all parameters should be given? In caption mention all of the involved pare metric values for which graphs are plotted, it will help to the reader compare the parametric effects for considered values of the parameters.

6. Conclusion should be precise.

7. What specific improvements should the authors consider regarding the

methodology? What further controls should be considered?

8. At the end novelty of the work should be highlighted clearly.

Reviewer #2: The study's theoretical foundation is not well defined or stated. The conceptual framework and theoretical presumptions that underpin the authors' analysis should be spelt out in detail, providing a more solid theoretical basis for the research.

Reviewer #3: The paper presents an interesting analysis of China’s rural financial risk, focusing on its regional differences, spatial distribution, and dynamic evolution, based on data from 2009 to 2021, covering 30 Chinese provinces. The literature review is relevant and well conducted, the research results are clearly presented, and the topic is interesting for the journal’s audience. However, the credibility of the findings is affected by the fact that the data underlying the conclusions have not been provided by the authors. In my opinion, there are some issues that need to be addressed and improved.

1. Please explain all the symbols used in the equations of section 3. In the present form, the equations are difficult to follow and understand on first reading.

2. I noticed that indicators x4, x9 and x13 are missing from the analysis in Table 2 and if I understood correctly from the explanations following Table 2, these indicators did not exhibit varying weight proportions within the rural financial risk index across the 30 analyzed provinces. However, it is not clear whether these indicators were taken into account in the calculation of the rural financial risk index? Were they relevant for this calculation? Please explain.

3. What are the limitations of the study? Provide a brief description of these limitations before the conclusions section.

4. According to the journal’s publication criteria, authors must follow standards and practices for the availability of data. According to your submission letter, “all data is fully available without restriction” in the manuscript and its Supporting Information files. However, I was unable to find any available data that could support the results and conclusions of the paper based on the indications of the manuscript. Please refer to the PLOS Data Policy and address this issue, as it is very important to comply with the journal’s publication criteria.

6. PLOS authors have the option to publish the peer review history of their article (what does this mean?). If published, this will include your full peer review and any attached files.

Reviewer #1: No

Reviewer #2: No

Reviewer #3: No

---

## [Author Response · Author response to Decision Letter 0]

8 Mar 2024

Thank you for giving us the opportunity to revise our article. The comments raised during this review highlighted quite a few areas in need of revision. We have taken every effort to address the comments, which has enabled us to significantly improve the contributions and delivery of our paper.

We have reproduced your comments (in blue font type) so that they can be considered in juxtaposition with our responses. Please refer to the document named Response to Reviewers.

---

## [Decision Letter · Decision Letter 1]

27 Mar 2024

Study on spatial distribution, regional differences and dynamic evolution of rural financial risk in China

PONE-D-23-36724R1

Dear Dr. He,

We’re pleased to inform you that your manuscript has been judged scientifically suitable for publication and will be formally accepted for publication once it meets all outstanding technical requirements.

Kind regards,

Dr. Walid Al-Shaar

Academic Editor

PLOS ONE

**Additional Editor Comments:**

The authors have successfully addressed all the reviewers' comments, and their manuscript has been accepted for publication in the journal. 

**Reviewers' comments:**

Reviewer #1: The authors have addressed all the issues properly, therefore, it is recommended for publication in its present form.

Reviewer #2: The research is valuable for understanding the challenges and opportunities in China's rural financial sector. The study's methodology and data analysis are thorough, providing a solid foundation for the conclusions drawn.

Reviewer #3: I consider the authors have addressed my comments in an acceptable manner. The article now meets the standards set by the journal.

---

## [Editor Report · Acceptance letter]

8 May 2024

PONE-D-23-36724R1 

PLOS ONE

Dear Dr. He, 

I'm pleased to inform you that your manuscript has been deemed suitable for publication in PLOS ONE. Congratulations! Your manuscript is now being handed over to our production team.

Kind regards, 

on behalf of

Dr. Walid Al-Shaar 

Academic Editor

PLOS ONE